# Identification of Clinical and Socioeconomic Predictors of Adjuvant Therapy after Trans-Oral Robotic Surgery in Patients with Oropharyngeal Squamous Cell Carcinoma

**DOI:** 10.3390/cancers12092474

**Published:** 2020-09-01

**Authors:** Sujith Baliga, Brett Klamer, Sachin Jhawar, Mauricio Gamez, Darrion Mitchell, Adriana Blakaj, John Grecula, Ulysses Gardner, Khaled Dibs, Matthew Old, Nolan Seim, Stephen Kang, Ricardo Carrau, Amit Agrawal, Vidhya Karivedu, Priyanka Bhateja, Enver Ozer, James Rocco, Marcelo Bonomi, Dukagjin Blakaj

**Affiliations:** 1Department of Radiation Oncology, The Ohio State University Wexner Medical Center, Columbus, OH 43210, USA; sujith.baliga@osumc.edu (S.B.); brett.klamer@osumc.edu (B.K.); sachin.jhawar@osumc.edu (S.J.); mauricio.gamez@osumc.edu (M.G.); darrion.mitchell@osumc.edu (D.M.); john.grecula@osumc.edu (J.G.); Khaled.dibs2@osumc.edu (K.D.); 2Department of Therapeutic Radiology, Yale School of Medicine, New Haven, CO 06520, USA; Adriana.Blakaj@yale.edu; 3Department of Medicine, Wright State University, Dayton, OH 45435, USA; gardner.89@wright.edu; 4Department of Otolaryngology, The Ohio State University Wexner Medical Center, Columbus, OH 43210, USA; matthew.old@osumc.edu (M.O.); nolan.seim@osumc.edu (N.S.); Stephen.kang@osumc.edu (S.K.); ricardo.carrau@osumc.edu (R.C.); amit.agrawal@osumc.edu (A.A.); enver.ozer@osumc.edu (E.O.); james.rocco@osumc.edu (J.R.); 5Department of Medical Oncology, The Ohio State University Wexner Medical Center, Columbus, OH 43210, USA; vidhya.karivedu@osumc.edu (V.K.); priyanka.bhateja@osumc.edu (P.B.); Marcelo.Bonomi@osumc.edu (M.B.)

**Keywords:** TORS, adjuvant therapy, radiation therapy, chemotherapy

## Abstract

**Simple Summary:**

Treatment of oropharynx cancers usually requires radiation, chemotherapy, surgery, or a combination of all three. Although these treatments are effective, they can cause both short- and long-term side effects, particularly when more than one treatment option is used. Robotic surgery is now an option for patients with oropharynx cancers, but it is not clear how many patients will require additional treatment with radiation, or combined chemotherapy and radiation, after surgical treatment. In this study we used a large national database of oropharynx cancer patients and found that two-thirds of patients who were treated with robotic surgery required radiation therapy and one-third required chemotherapy with radiation. In addition, we found that the true tumor stage of the patients in this study was often higher than was initially thought prior to surgery. Finally, patients treated at high volume surgical centers were more likely to have more of their tumor removed compared to those at low volume facilities. Better survival quality of oropharynx cancer patients could be achieved by improving pre-surgical selection of patients so that the number of treatment modalities is reduced.

**Abstract:**

Trans-oral robotic surgery (TORS) has emerged as an important surgical treatment option in the management of human papillomavirus (HPV)-positive and -negative oropharynx cancer. However, treatment selection is paramount to ensure that patients will not require multimodality adjuvant therapy. In this study, we determined predictors of adjuvant therapy in TORS-treated patients. The National Cancer Database (NCDB) was used to identify patients with newly diagnosed clinical T1-T4, N0-N3 oropharyngeal squamous cell carcinoma who underwent TORS between 2010–2016. Kaplan–Meier survival analysis was used to estimate overall survival (OS). A total of 2999 patients were studied, and the five-year OS for the entire cohort was 82.5%, and for HPV-positive and -negative cohorts it was 88.3% and 67.9%, respectively (*p* < 0.001). Among all patients treated with TORS, 35.1% of patients received no additional treatment, 33.5% received adjuvant radiation alone (RT), and 31.3% received adjuvant chemoradiation. The N stage was pathologically upstaged in 629 (20.9%) patients after TORS. Patients treated at higher-volume centers were more likely to have negative surgical margins (OR: 0.96, 95% CI: 0.94, 0.98, *p* < 0.001), but this did not influence the receipt of adjuvant therapy. The high rate of adjuvant multimodality treatment after TORS suggests a need for improved patient selection. Limitations of this study, including lack of data on loco-regional control, progression free survival, acute and late toxicities, and utilization of pretreatment PET/CT imaging, should be addressed in future studies.

## 1. Introduction

The incidence of oropharynx cancers has steadily increased in the last decade, likely driven by a concomitant rise in the incidence of human papillomavirus (HPV)-associated oropharyngeal squamous cell carcinoma (OPSCC) [1]. Several landmark studies have demonstrated superior survival outcomes for HPV-associated OPSCC compared to tobacco- and alcohol-driven OPSCC [2,3]. As outcomes continue to improve in OPSCC, de-intensification of treatment has emerged as an attractive strategy in this favorable subset of patients. Several prospective trials over the last decade have assessed de-escalation of radiotherapy (RT) dose in both the definitive and adjuvant settings. TransOral Robotic Surgery (TORS) is a promising, minimally invasive surgical strategy for treatment de-intensification in the management of OPSCC. Recent studies have shown excellent quality-of-life, functional, and oncologic outcomes in patients treated with TORS [4,5]. The primary concern regarding the use of TORS in OPSCC is appropriate patient selection to avoid multimodality therapy, which may result in increased toxicity. Previously, we demonstrated excellent overall survival (OS), but high rates of adjuvant therapy for patients undergoing TORS for T1-T2, N0-N2b OPSCC treated between 2010–2014 [6]. As data from prospective de-escalation trials emerge, we hypothesize that better patient selection should translate into a reduction in the need for adjuvant therapy after TORS. The objective of this study was to address the impact of institutional volume and patient selection on the need for adjuvant therapy in TORS-treated patients. 

## 2. Results

### 2.1. Patient Characteristics

We identified 4954 nonmetastatic OPSCC patients who were treated with TORS from 2010–2016, which formed our initial data set. After applying the exclusion criteria previously described, 1955 patients were excluded, with 2999 patients included in the final analysis. A flow diagram of patient selection criteria is shown in Figure 1.

The baseline pretreatment demographic, socioeconomic, and tumor characteristics are shown in Table 1. The majority of patients were clinically staged as T1-T2 (*n* = 2795, 93.2%). Of all patients in the cohort, 1368 patients (45.6%) were American Joint Committee on Cancer (AJCC ) seventh edition T1-2 N0-N1, 1427 (47.6%) were T1-T2 N2a-N3, 90 (3.0%) were T3-T4 N0-N1, and 114 (3.8%) were T3-T4 N2a-N3. Only 42 (1.4%) patients had N3 disease. Of the entire cohort, 2076 (69.2%) were HPV positive, 563 (18.8%) were HPV negative, and 360 (12.0%) had HPV status missing. Of patients in the cohort, 2704 (90.2%) were Caucasian, 1854 (61.8%) had private insurance, and 66.2% of patients were in the top 50% of the income quartile. Of note, 2905 (96.9%) patients underwent a lymph node dissection with a median of 31 lymph nodes examined and a median of one positive node.

### 2.2. Survival Analysis

Of the 2999 patients, 2391 (79.7%) were included in the overall survival analysis and 608 (20.3%) had missing follow-up times. The five-year OS for the entire cohort was 82.5% (95% Confidence Interval (CI): 80.5%, 84.6%) (Figure 2A). The five-year OS for HPV-positive versus HPV-negative patients was 88.3% (95% CI: 86.1%, 90.5%) versus 67.9% (95% CI: 61.7%, 74.6%), respectively (Figure 2B). Patients with negative surgical margins had a five-year OS of 83.7% (95% CI: 81.5%, 85.9%) versus 76.0% (95% CI: 70.4%, 82.0%) for positive surgical margins (Figure 2C). 

On univariate analysis, the receipt of adjuvant radiation therapy (Hazard ratio: (HR): 0.65, 95% CI: 0.48, 0.87), adjuvant chemoradiation (HR: 0.81, 95% CI: 0.62, 1.07), HPV status (HR: 0.31, 95% CI: 0.24, 0.42), Charlson comorbidity score (HR: 1.56, 95% CI: 1.18, 2.0), clinical Tumor (T) stage (T3/4 vs. T1, HR: 1.75, 95% CI:1.16, 2.64), clinical N stage (N2c+ vs. N0, HR: 1.29, 95% CI: 0.8, 2.07), surgical margin status (positive vs. negative, HR: 1.38, 95% CI: 1.03, 1.85), Extracapsular extension (ECE) (positive ECE vs. negative, HR: 1.57, 95% CI: 1.18, 2.09), Lymphovascular space invasion (LVSI) (positive LVSI vs. negative, HR: 1.76, 95% CI: 1.36, 2.28), and radiation dose (HR: 0.99, 95% CI: 0.99, 1) were all independently associated with overall survival.

On multivariate analysis, Medicare (HR: 2.46, 95% CI: 1.66, 3.64, *p* < 0.001) or Medicaid insurance status (HR: 1.99, 95% CI: 1.11, 3.57, *p* = 0.021), Charlson comorbidity score ≥2 (HR: 3.2, 95% CI: 1.91, 5.35, *p* < 0.001), presence of ECE (HR: 1.6, 95% CI: 1.08, 2.37, *p* = 0.019), and presence of LVSI (HR: 1.88, 95% CI: 1.35, 2.6, *p* < 0.001) were associated with inferior OS. Conversely, HPV-positive status (HR: 0.3, 95% CI: 0.22, 0.41, *p* < 0.001) was associated with improved OS (Table 2, Figure 3).

### 2.3. Utilization/Predictors of Adjuvant Treatment 

The use of TORS increased 292% between 2010 and 2016, from 208 patients in 2010 to 608 patients in 2016 (Figure 4A). Among all patients treated with TORS, 35.1% of patients received no additional treatment, 33.5% required adjuvant RT, and 31.3% required adjuvant chemoradiation (chemoRT) (Table 1). Among patients who were clinical (c) T1-2, N0-1, 714 (53.1%) received no additional therapy, 398 (29.1%) underwent adjuvant RT, and 243 (17.8%) required adjuvant chemoRT. For patients who were T3-T4, N2-N3, 14 (12.3%) received no additional treatment (reasons for omission of treatment are not provided in NCDB), 38 (33.3%) underwent adjuvant RT, and 62 (54.4%) patients received adjuvant chemoRT. The proportion of TORS patients who received no additional treatment and adjuvant RT alone increased from 2010 to 2016, while the proportion of patients who received adjuvant chemoRT decreased in that same time period (Figure 4B). The proportion of patients with T1/T2, N0/N1 disease who did not receive any additional treatment increased from 2010-2016, while the proportion of patients who received adjuvant RT who had T1/T2, N2/N3 disease increased over the same time period (Figure 4C). Interestingly, the proportion of patients who received adjuvant chemoradiation who were T3/T4, N0/N1 and T3/T4, N2/N3 sharply decreased from 2010–2016 (Figure 4C).

Patients treated with TORS at higher-volume centers were less likely to have positive margins (Odds Ratio (OR): 0.96, 95% CI: 0.94, 0.98, *p* < 0.001) compared to those treated at a low-volume center. For every one patient increase in mean volume, we expect to see a 4% decrease in the odds of a positive margin. The relationship between mean volume of cases and center frequency is shown in Appendix A. There was no relationship between center volume and the receipt of adjuvant therapy (OR: 1.01, 95% CI: 0.99, 1.03, *p* = 0.242).

We identified clinical, pathologic, and socioeconomic predictors for use of adjuvant therapy in TORS-treated patients. Adjusting for sociodemographic variables only, clinical N stage, clinical T stage, overall TNM Classification of Malignant Tumors (TNM) stage, surgical margins, base of tongue subsite, ECE status, HPV status, volume of cases, and year of diagnosis were associated with the type of adjuvant therapy received (likelihood ratio test (LRT) *p* < 0.05 for each).

The multivariable proportional odds’ model showed a decrease in odds of adjuvant therapy as diagnosis year increased (OR: 0.93, 95% CI: 0.89, 0.96). Compared to negative ECE, patients with positive ECE was more likely to receive adjuvant therapy (OR: 6.06, 95% CI: 5.03, 7.3). Positive margins were associated with increased likelihood of adjuvant therapy (OR: 2.28, 95% CI: 1.81, 2.88), as was positive LVSI (OR: 1.56, 95% CI: 1.3, 1.88). Patients with higher-stage disease were more likely to receive adjuvant therapy (Figure 5). 

Next, we aimed to identify how the clinical and pathologic stage changed within this cohort after TORS. The N stage was pathologically upstaged in 629 (20.9%) patients (Figure 6A). The T stage was upstaged pathologically in 530 (15.8%) patients (Figure 6B). Of all clinically node-negative patients, 205 (25.3%) were upstaged to pathologically node-positive. The greatest change in stage occurred in patients who were clinically N1, of which 368 (50.3%) were upstaged to N2 or greater (Figure 6A). Pathologic downstaging was uncommon, with the N stage down staged in only 218 (7.2%) of patients and the T stage down staged in 324 (10.8%) patients.

### 2.4. Predictors of Radiation Dose 

Of the entire cohort, 64.9% of patients received adjuvant RT and were treated to a median dose of 60 Gy (Interquartile range (IQR): 59.4–64 Gy). A total of 1651 patients received radiation doses between 50 Gy and 70 Gy and the association between dose and explanatory variables for these patients was explored. Adjusting for sociodemographic variables, higher clinical T stage, higher clinical N stage, positive surgical margins, ECE, and receipt of chemotherapy were all factors associated with increased likelihood of receiving a higher radiation dose. 

In the multivariate model, Hispanic race (3.85 Gy, 95% CI: 0.42, 7.29), Medicare insurance (1.56 Gy, 95% CI: 0.06, 3.06), receipt of chemotherapy (4.08 Gy, 95% CI: 2.75, 5.41), positive ECE (1.46 Gy, 95% CI: 0.19, 2.73), and positive surgical margins (2.62 Gy, 95% CI: 1.12, 4.11) were associated with higher radiation dose (Appendix A). Patients who were ECE-positive received a mean total dose of 3 Gy higher than those who were ECE-negative. 

## 3. Discussion

To our knowledge, this is the largest national retrospective database study of TORS-treated patients and describes in detail the patterns and predictors of adjuvant therapy. Survival outcomes for TORS-treated patients in our study were comparable to those published in the literature, both for the overall cohort and for the HPV-positive subgroup [2,7]. While we have previously shown high rates of adjuvant therapy in a prior study of TORS-treated patients [6], this study included an additional 319 patients and revealed several novel findings not previously known, including the impact of socioeconomic status, race, discordance between clinical and pathologic stage, and center volume as factors influencing the necessity for adjuvant treatment. 

Our study demonstrates a high rate of adjuvant therapy after TORS (64.9%), consistent with the results of previously published retrospective studies, which have reported adjuvant treatment in 57–88% of patients [8,9,10,11,12,13]. While retrospective studies are limited by selection bias, the recently prospective ORATOR trial [14] reported that 47% of patients required adjuvant RT alone and 24% of TORS-treated patients required adjuvant chemoradiation. These findings underscore the importance of appropriate patient selection from TORS, given the increased toxicity associated with multimodality therapy if it is subsequently required after TORS. While we were encouraged to see that treatment with TORS resulted in the elimination of RT in 33% of patients, it is clear that even for patients with small HPV-positive tumors, a considerable percentage will require adjuvant treatment. The accurate identification of patients with low-volume (pT1-2) and negative-nodal disease with the lack of high-risk features (perineural invasion (PNI), LVSI, positive margins) is challenging and continues to remain the major barrier to employing TORS in this patient population. 

Even among the most favorable subgroup of TORS-treated patients in our study (cT1-2, N0-N1), approximately 47% required adjuvant therapy and 18% required both chemotherapy and RT. This has traditionally been a subgroup that has been treated with definitive radiation alone with good oncologic outcomes and only 10–15% requiring a salvage neck dissection after. Therefore, the increase in the number of treatment modalities is likely to also increase long-term toxicities and decrease quality of life in these patients. The etiology for the high rate of adjuvant therapy in this subgroup is unclear. However, our finding that over 50% of patients who were clinically staged as N1 were pathologically upstaged to N2 or greater may account, in part, for the high rate of adjuvant therapy. Gildener-Leapman et al. [15] performed a retrospective study of TORS-treated patients showing that neck dissection upstaged the nodal status in 36% of patients. McMullen et al. [16] retrospectively reviewed the incidence of occult nodal disease in patients undergoing TORS with neck dissection and demonstrated that 22% of patients were pathologically upstaged from N0/N1 to N2 or greater. Our study provides some encouraging evidence of improved patient selection, with the rate of adjuvant chemoradiation dropping from over 40% in 2010 to less than 30 percent in 2016 (Figure 4B). Nevertheless, pathologic upstaging is likely to continue to be a significant barrier to reducing the number of adjuvant treatments in this cohort of patients. Careful review of preoperative imaging and incorporation of PET/CT imaging may improve our ability to detect occult nodal disease and should continue to be explored as a strategy to more accurately stage these patients and their optimal selection for TORS treatment. In addition, artificial and deep learning algorithms have been shown to have excellent accuracy at predicting extra-nodal extension in two external data sets in Head and Neck Squamous Cell Carcinoma (HNSCC) patients and represent a promising tool to improve patient selection for TORS [17]. In this study, the deep learning algorithm was better able to reliably predict ECE compared to radiology review.

The rate of positive surgical margins in our study was 13.5%. This is higher than was previously reported in a systematic review, which demonstrated that the overall percent of positive surgical margins was 7.8% [18]. TORS should continue to be performed at high-volume centers, and our study demonstrates that treatment at a high-volume center is associated with a decreased risk of positive margins. These results are consistent with prior retrospective studies using the National Cancer Database (NCDB) [19]. In addition, it is encouraging to see that the use of adjuvant therapy decreased as the year of diagnosis increased, possibly a result of increased surgeon comfort with the technique and the robotic device. 

An interesting finding in this study was the negative impact of insurance status on survival, even after adjusting for socioeconomic status and cancer stage. This has been previously reported in other head and neck cohorts, but is not well described in the TORS-treated population [20]. Patients with Medicare or Medicaid insurance had worse OS compared to those with private insurance. While it could be argued that patients with Medicare insurance are, by definition, older and may have other co-existing comorbidities, performance status was accounted for in the multivariable model. Another potential explanation is that private insurance status could reflect a patient population with a higher socioeconomic status and, therefore, may be more willing to seek treatment at high-volume centers. This should be further explored in future studies. 

Another important distinction to highlight is that, although our study tabulated the percentage of patients who received RT, we did not delineate which patients appropriately received adjuvant therapy based on national guidelines. A recent NCDB study by Bates et al. [21] demonstrated that nearly 20% and 33% of patients with an indication for adjuvant RT or adjuvant chemoradiation, respectively, failed to receive the appropriate treatment. Therefore, the rates of appropriate adjuvant treatment may be higher than reported in our study.

The finding that being of Hispanic race is associated with higher adjuvant RT dose was unexpected and deserves further exploration in other data sets. One explanation could be that Hispanic patients have a higher burden and volume of disease at initial presentation, resulting in a higher rate of extracapsular extension or positive margins. Whether this is due to a more aggressive biology of their cancer or reflects socioeconomic disparities should be explored in future studies. A previous study of 49 Hispanic head and neck cancer patients demonstrated a trend toward a more advanced stage of cancer, but no impact on OS [22]. A case-matched analysis of HNSCC patients by race performed by Schrank et al. [23] demonstrated a higher rate of radiotherapy use among Hispanics and African American patients compared to Caucasian patients.

There are several limitations associated with this study. The NCDB does not provide data for local control, progression-free survival, or toxicity, which limited our ability to compare these outcomes to a nonsurgical treatment approach. Outcomes regarding long-term toxicities are also unavailable, which limited our ability to assess the impact of multimodality therapy on quality of life measures. Data regarding the utilization of PET/CT pretreatment imaging is not readily available in the NCDB, limiting our ability to stratify adjuvant treatment by the use of imaging. Next, our study aggregated adjuvant data from patients in all clinical stage groups, which may show a higher rate of adjuvant therapy. We felt it was important to include all TORS-treated patients to illuminate the current patterns of care in TORS usage. However, we were reassured to see the low rates of TORS utilization among T3-T4/N2-N3 patients, and these patients should continue to be treated with either surgery followed by chemoradiation/RT or primary chemoradiation as the standard of care. In well-selected patients who have good performance status, surgery followed by chemoradiation has been shown to have improved disease-specific survival compared to chemoradiation [24]. Finally, information regarding smoking status is not available in the NCDB, which is a major prognostic factor for outcomes in both HPV-positive and -negative subgroups. 

For patients who undergo TORS, de-intensification or elimination of adjuvant therapies should continue to be the primary aim. The results of Eastern Cooperative Oncology Group (ECOG) 3311 were presented in abstract form at the American Society of Clinical Oncology (ASCO) Annual Meeting of 2020 and demonstrated favorable disease outcome data, particularly in the intermediate risk group, where RT dose de-escalation to 50 Gy was associated with a two-year progression free survival (PFS) of 95%. Although the previous data is encouraging, for patients with early stage disease, the long-term toxicity benefit of surgery followed by adjuvant RT (either to 50 or 60 Gy) compared to primary chemoradiation to 70 Gy has not been established. While the ORATOR trial showed superior swallow QOL scores at one-year posttreatment, the group that received adjuvant RT after TORS received at least 60 Gy [14]. Longer-term follow-up will be needed to understand the benefits in terms of radiation dose reduction in long-term toxicity in the TORS-treated subgroup. 

Of note, in ECOG 3311, TORS was performed at high-volume centers and required credentialing of surgeons as part of site participation. Overall outcomes in these trials are encouraging, but long-term data are required to assess outcomes at five years. It is crucial that initial patient selection of TORS patients be critically assessed in order to avoid surgery in patients who are more likely to have high-risk features after surgery and, thus, patients who would subsequently require radiation or chemoradiation. 

In summary, this is the largest retrospective study of primary TORS-treated patients in OPSCC and demonstrates that the utilization of TORS has significantly increased from 2010–2016. However, the high percentage of patients requiring adjuvant therapy, particularly those who receive multimodality therapy after TORS, suggests that improved presurgical patient selection is needed in order to identify patients at high risk of having positive margins and ECE. Nevertheless, the TORS approach remains a promising strategy in OPSCC and should continue to be evaluated in prospective clinical trials. 

## 4. Materials and Methods

The National Cancer Database (NCDB) is a joint project of the Commission on Cancer (CoC) of the American College of Surgeons and the American Cancer Society. The CoC’s NCDB and the participating hospitals were the sources of the de-identified data used herein; they did not verify and are not responsible for the statistical validity of the data analysis or the conclusions derived by the authors. Data within the NCDB include basic demographics, cancer staging, comorbidities, therapies delivered during the first course of treatment, and OS. The NCDB does not capture disease recurrence or salvage therapies.

The NCDB was queried to identify patients with newly diagnosed AJCC seventh edition clinical T1-T4, N0-N3 OPSCC treated with TORS between 2010–2016. We included patients who were ≥18 years of age with a diagnosis of squamous cell carcinoma of the base of tongue or tonsil. Patients with any non-metastatic stage were included in the analysis. Primary site codes for tonsil were “C024”, “C090”, “C091”, “C098”, and “C099”. The primary site code for base of tongue was “C019”. Determination of the use of TORS was based on surgical approach being categorized as “Robotic”. Patients were excluded for the following reasons: (1) Unknown clinical T or N stage, (2) unknown pathologic T or N stage, (3) the presence of metastatic disease; (4) if surgical margin status was unavailable, (5) if radiation was given to an area other than the head and neck, (6) if details regarding the administration of chemotherapy were unavailable, and (7) to ensure that the selected patients had a true oncologic resection, we excluded patients who had a biopsy alone. We also included patients only with surveillance, epidemiology, and end results (SEER) surgery codes that indicated an oncologic resection (pharyngectomy or radical pharyngectomy), which included 30, 31, 32, 40, 41, 42, 50, 51, 52, and 90. A patient was considered to have a negative margin if all margins were grossly and microscopically negative and there was no tumor on the inked margin of resection. A margin was considered positive if there was microscopic or macroscopic residual tumor. 

### Statistical Analysis

Patient characteristics were summarized using frequency and percent for categorical variables and the median and interquartile range for numeric variables. Categorical variables were compared with McNemar’s test for paired data. Center volume was defined as the mean number of cases treated per year. The univariable relationship between center volume and the binary outcome variables’ adjuvant status and surgical margin were assessed with logistic mixed models. OS was defined as the time, in months, between the date of diagnosis and the date of death, and censored at the date of last follow-up for those still alive. Hazard ratios and their two-sided confidence intervals were calculated using Cox proportional hazards regression models. Adjuvant therapy was modeled using proportional odds regression models. Relationships with radiation dose (50–70 Gy) were summarized using truncated linear regression models. All regression models utilized complete case analysis. Potential explanatory variables were selected based on literature review and clinical experience. Explanatory variables included in the final multivariable regression models were chosen based on prespecified clinical significance (HPV status, adjuvant status, surgical margins) and the reliability of each variable’s importance ranking, based on its fraction of explainable outcome variation (partial chi-square), across bootstrap re-samples. Among the set of sociodemographic explanatory variables (age, sex, race, insurance status, and median income) the least reliable variable was removed from each final multivariable model. Among the set of clinical explanatory variables (extracapsular extension (ECE), lymphovascular space invasion (LVSI), T stage, N stage, overall stage, Charlson comorbidity score), those which had nonsignificant partial chi-square values or poor reliability were removed from the model. A two-sided *p*-value of less than 0.05 was considered to indicate statistical significance. Statistical analysis was conducted using R (version 3.6.3) with the survival (version 3.1-12), rms (version 5.1-4), and lme4 (version 1.1-23) packages.

## 5. Conclusions

In conclusion, our study shows that TORS is becoming more prevalent nationally in the treatment of OPSCC. However, the use of TORS also is associated with a higher likelihood of adjuvant treatment (either RT alone or chemoRT). Better patient selection through improved imaging may allow us to differentiate a subset of patients unsuitable for TORS, particularly for those with extra-nodal extension, thereby sparing the toxic effects of multimodality treatment. 

## Figures and Tables

**Figure 1 cancers-12-02474-f001:**
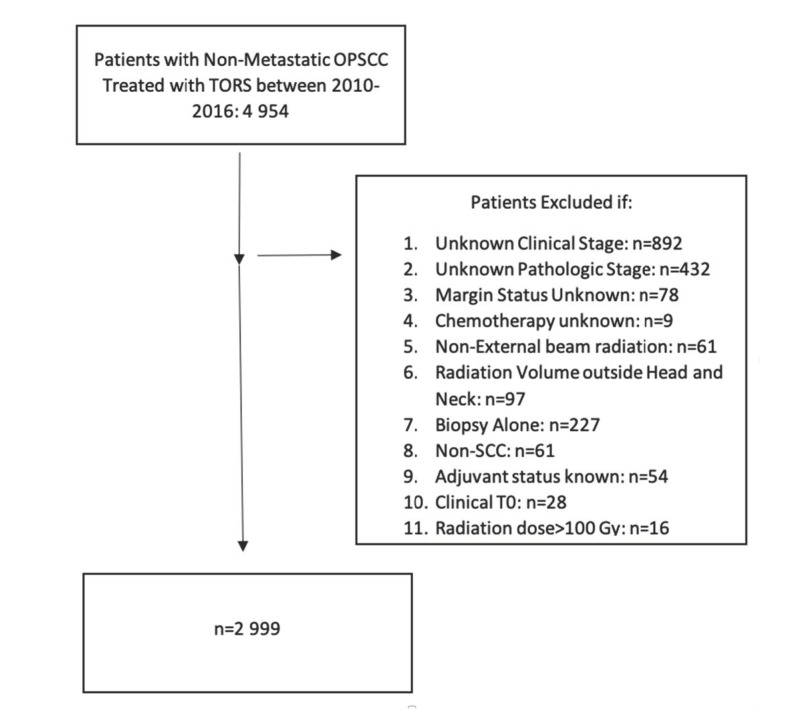
Selection criteria.

**Figure 2 cancers-12-02474-f002:**
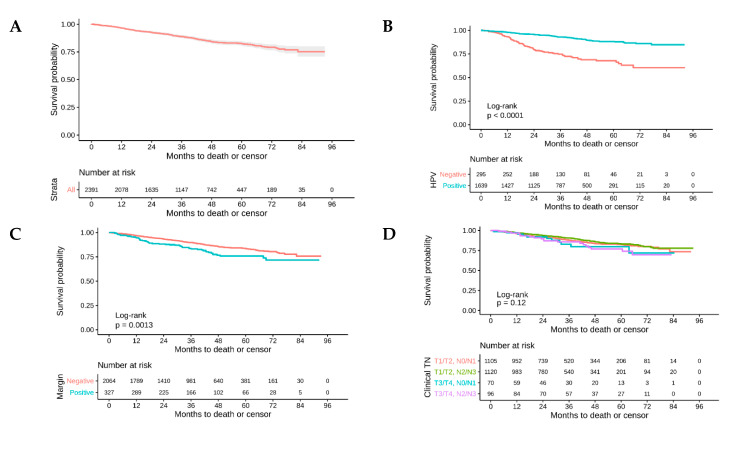
Overall survival in Trans-Oral Robotic Surgery (TORS)-treated patients for (**A**) the entire cohort, (**B**) by human papillomavirus (HPV) status, (**C**) by surgical margin status, and (**D**) by stage.

**Figure 3 cancers-12-02474-f003:**
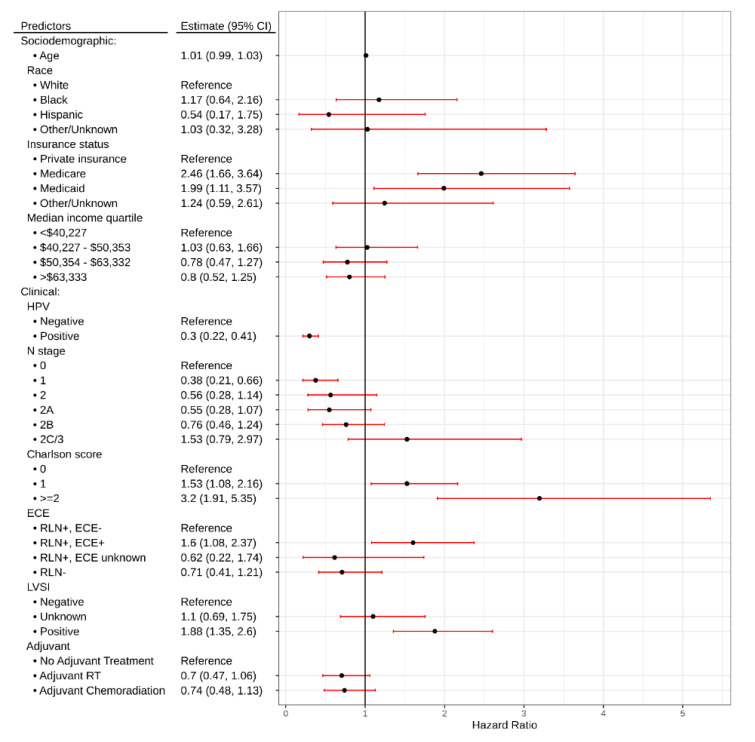
Forrest plot demonstrating hazard ratios for variables associated with survival in Oropharyngeal Squamous Cell Carcinoma patients treated with TORS.

**Figure 4 cancers-12-02474-f004:**
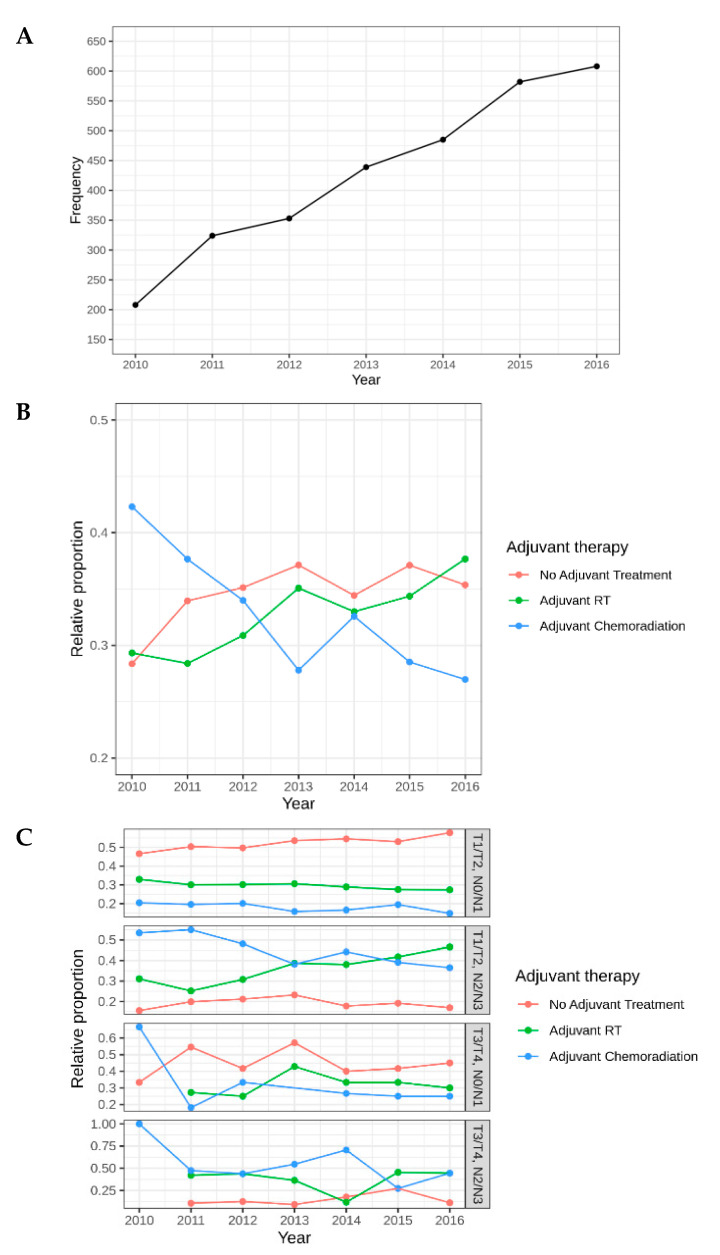
(**A**) Overall frequency of TORS use, (**B)** relative proportion of adjuvant therapy over time, (**C**) use of adjuvant therapy by Tumor (T) and Node (N) stage.

**Figure 5 cancers-12-02474-f005:**
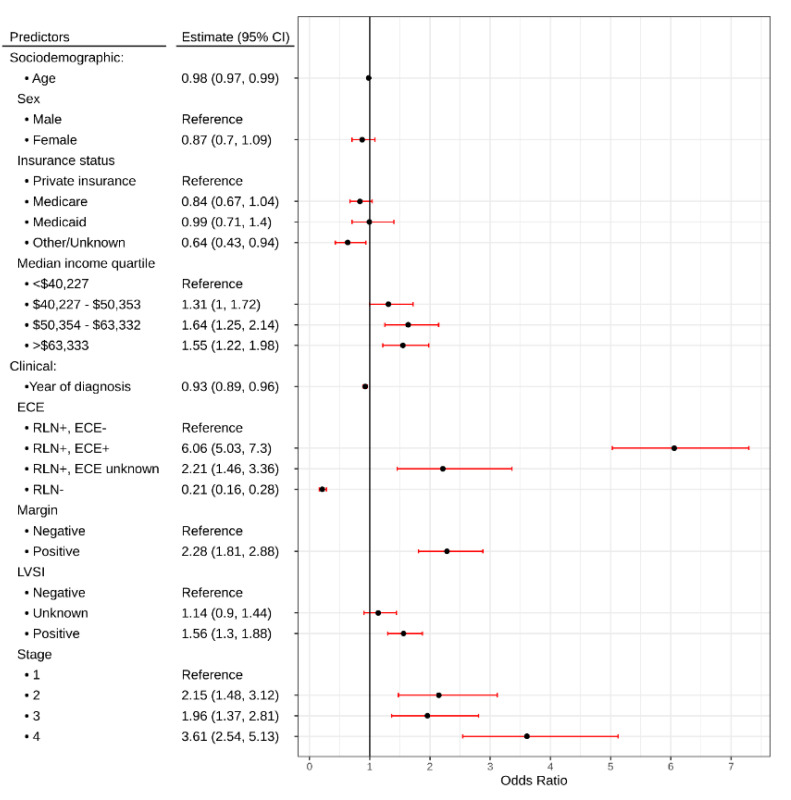
Forrest plot depicting odds ratios for receipt of adjuvant therapy in TORS-treated patients. RLN: Regional lymph node; ECE: Extracapsular extension; LVSI: Lymphovascular space invasion.

**Figure 6 cancers-12-02474-f006:**
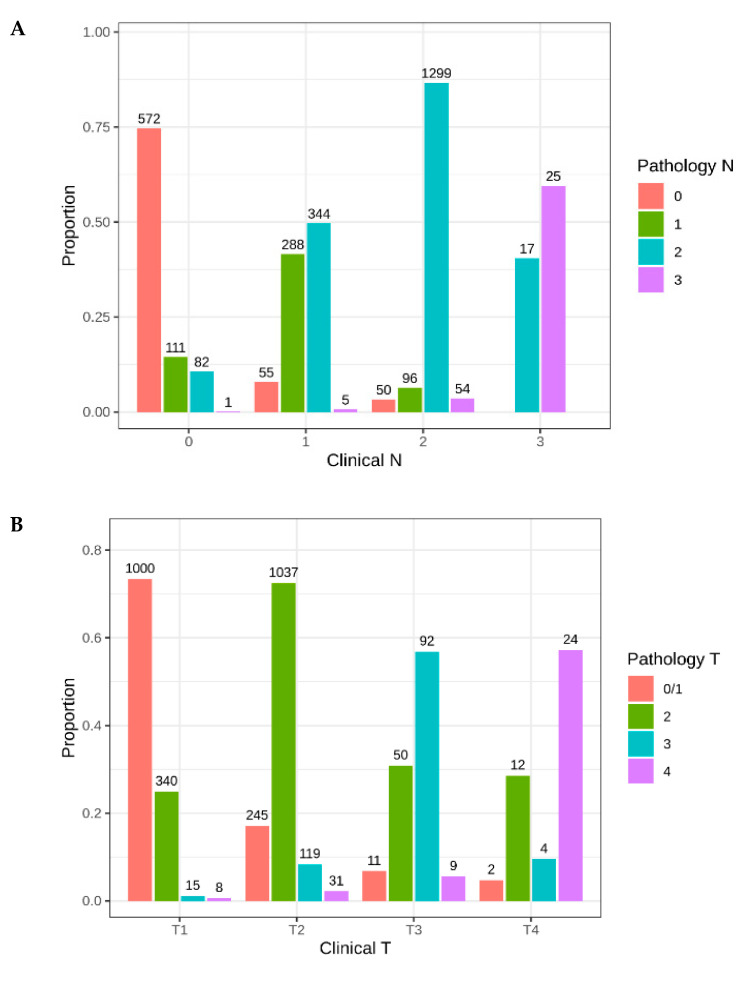
Change in (**A**) pathological N and (**B**) T stage after TORS surgery.

**Table 1 cancers-12-02474-t001:** Patient characteristics.

Patient Characteristics	Total Number (%)
**Total**	**2999**
Age (years)	
Median	59
IQR Range	53–66
Sex	
Male	2550 (85%)
Female	449 (15%)
Race	
White	2704 (90.2%)
Black	136 (4.5%)
Hispanic	83 (2.8%)
Other/Unknown	76 (2.5%)
Hispanic	
No	2866 (95.6%)
Yes	83 (2.8%)
Unknown	50 (1.7%)
Comorbidity Score	
0	2367 (78.9%)
1	495 (16.5%)
≥2	137 (4.6%)
Subsite	
Tonsil	1937 (64.6%)
Base of Tongue	1062 (35.4%)
HPV Status	
Positive	2076 (69.2%)
Negative	563 (18.8%)
Unknown	360 (12.0%)
Clinical T Stage AJCC 7th edition	
T1	1363 (45.4%)
T2	1432 (47.7%)
T3	162 (5.4%)
T4	42 (1.4%)
Clinical N Stage AJCC 7th edition	
N0	766 (25.5%)
N1	692 (23.1%)
N2*	212 (7.1%)
N2a	331 (11%)
N2b	862 (28.7%)
N2c	94 (3.3%)
N3	42 (1.4%)
Surgical Margin Status	
Negative	2595 (86.5%)
Positive	404 (13.5%)
LVSI	
Negative	1911 (63.7%)
Positive	708 (23.6%)
Unknown	380 (12.7%)
Extracapsular Extension (ECE)	
Yes	840 (28.0%)
No	2049 (68.3%)
Unknown	110 (3.7%)
Adjuvant Therapy	
None	1054 (35.1%)
Radiation Alone	1005 (33.5%)
Chemotherapy and Radiation	940 (31.3%)
Insurance Status	
Private	1854 (61.8%)
Medicare	852 (28.4%)
Medicaid	162 (5.4%)
Other Government	62 (2.1%)
Not Insured	48 (1.6%)
Unknown	21 (0.7%)
Facility	
Academic	2512 (83.8%)
Community Center	320 (9.8%)
Integrated Network	157 (5.2%)
Unknown	35 (1.2%)
Median Income Quartiles	
<40,227	390 (13.0%)
40,227–50,353	580 (19.3%)
50,354–63,332	648 (21.6%)
>63,333	1337 (44.6%)
Unknown	44 (1.5%)

Abbreviations: N/A: Not available, RT: Radiation therapy, Nx: unknown N-stage, LVSI: Lymphovascular space invasion. Data is presented as No. (% of the entire group). *A minority of TORS patients were classified as N2 and were not categorized as N2a-c. Data is presented as No. (% of the entire group).

**Table 2 cancers-12-02474-t002:** Multivariable analysis of predictors of Overall Survival (OS) in TORS-treated patients.

Predictors	Overall Survival		
	Hazard Ratio	CI	*p*
Age	1.01	0.99–1.03	0.32
Race			
White	Reference		
Black	1.17	0.64–2.16	0.608
Hispanic	0.54	0.17–1.75	0.307
Other/Unknown	1.03	0.32–3.28	0.962
Insurance status			
Private Insurance	Reference		
Medicare	2.46	1.66–3.64	**<0.001**
Medicaid	1.99	1.11–3.57	**0.021**
Other/Unknown	1.24	0.59–2.61	0.564
Median Income Quartile			
<$40,227	Reference		
$40,227–$50,353	1.03	0.63–1.66	0.919
$50,354–$63,332	0.78	0.47–1.27	0.314
>$63,333	0.8	0.52–1.25	0.325
HPV			
Negative	Reference		
Positive	0.3	0.22–0.41	**<0.001**
N Stage			
0	Reference		
1	0.38	0.21–0.66	**0.001**
2	0.56	0.28–1.14	0.113
2A	0.55	0.28–1.07	0.08
2B	0.76	0.46–1.24	0.271
2C/3	1.53	0.79–2.97	0.212
Charlson score			
0	Reference		
1	1.53	1.08 – 2.16	**0.018**
≥2	3.2	1.91–5.35	**<0.001**
ECE			
RLN+, ECE-	Reference		
RLN+, ECE+	1.6	1.08–2.37	**0.019**
RLN+, ECE unknown	0.62	0.22–1.74	0.36
RLN-	0.71	0.41–1.21	0.204
LVSI			
Negative	Reference		
Unknown	1.1	0.69–1.75	0.692
Positive	1.88	1.35–2.60	**<0.001**
Adjuvant			
No adjuvant treatment			
Adjuvant treatment	0.7	0.47–1.06	0.094
Adjuvant chemoradiation	0.74	0.48–1.13	0.158
Observations	1840		
R^2^ Nagelkerke	0.645		

RLN: Regional lymph node. ECE: Extracapsular extension. Bold *p*-values are statistically significant.

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
