# Peer review of "Identification of Clinical and Socioeconomic Predictors of Adjuvant Therapy after Trans-Oral Robotic Surgery in Patients with Oropharyngeal Squamous Cell Carcinoma"

_cancers, 2020, doi:10.3390/cancers12092474_

Round 1

Reviewer 1 Report

TORS is used more frequently for oropharyngeal cancer, however in, as also shown in this large database, 1/3 followed by radiation and in 1/3 even with chemoradiation. For HPV positive tumors and Stage I-II HPV negative tumors with (chemo) radiation the cure rates are high. So, the important question remains what TORS has to add for locoregional control and DFS. Besides cost and toxicity will increase by the inclusion of TORS in the treatment. The ORATOR study showed that swallowing complaints for low stage HPV+ were lower for the radiation group compared to the TORS group.

The authors choose for another point of view: how to better select patients for TORS, so adjuvant (chemo) radiation will only be indicated in a small group.

Unfortunately the database has many missing important factors, as data about toxicity, locoregional control and DFS. Only overall survival is recorded.

In this study with around 3000 patients 93% were T1-T2. 

TORS was combined with a neckdissection in 97%, 50% of the cN1 were upstaged to pN2 or pN3. 

The question how to better select the patients for TORS only was not answered in this study. Without data on locoregional control and DFS this question will be difficult to answer.

Why adjuvant treatment was indicated in 2/3 patients?

I: Upstaging of N-status by neckdissection. With primary (chemo) radiation in the vast majority these patients will be cured without need for a neckdissection. How often a FDG-PET CT was performed before TORS?

II: positive surgical margins, which may improve if TORS will only be used in high volume centers

III: The insurance status was an independent prognostic factor for OS. This is an interesting result of the study, but may tell more about the selection of the patients for TORS: more high income, more private insurance, better general condition?

IV: Hispanic patients receive a higher dose of radiation?? Its is hard to understand that radiation oncologist use this as a prognostic factor in the decision tree for the appropriate tumor dose in their cases.

So, in conclusion: this study was performed to come-up with guidelines how to select patient for TORS without the need for adjuvant treatment, but at the end no guidelines are or could be given. So, maybe we have to reconsider use TORS for these indications.

More emphasis on the shortcommings of the study and the remarks mentioned should be given in he discussion.

Reviewer 2 Report

  1. Surgical Margins: how do you define a negative margin?  e.g. no tumor on the ink? 1mm? 2mm? etc
  2. ~1/3 patients received RT. Does this include patients who had indications for RT and declined?
  3. In Section 2.4 you mention 64.9% of patient required RT. What is this "requirement" based on? 
  4. In Section 2.4 when you discuss "higher radiation doses" can you quantify this? For example: patients with ENE received an average of 68Gy, which was on average 10 Gy more than those who did not. 
  5. Figure 5 needs definitions of abbreviations. 
  6. Discussion:
    1. Good point about TORS being done at high volume centers. There are community "dabblers" out there who take a TORS course and start doing their own cases instead of referring them to high volume centers where multi-D groups focus on delivering high quality care. The idea that these patients need specialized care at high volume centers cannot be over emphasized. Do you have figure for # of cases per year that makes a high volume center?
    2. Insurance status: could it be possible that the private insurance patients have higher SES and seek out higher volume centers for their care? Maybe that has something to do with survival?
    3. T3/T4;N2-N3: you mention that CRT is and should be the standard of care. The NCCN guidelines suggest that CRT or surgery are appropriate treatments for these cases for p16neg and p16pos disease. High volume surgical centers have show improved survival with Surgery+RT/CRT vs CRT for High Staged disease (especially p16 neg).  e.g.:

      O'Connell D, Seikaly H, Murphy R, Fung C, Cooper T, Knox A, Scrimger R, Harris JR Primary surgery versus chemoradiotherapy for advanced oropharyngeal cancers: a longitudinal population study. J Otolaryngol Head Neck Surg. 2013 Apr 22;42(1):31.

      While most centers do CRT for these patients, surgery is a viable option with excellent outcomes at high volume centers. You may want to mention this as a possibility.
    4. Avoiding Multi-Modal Treatment: this is especially true for p16pos disease. We absolutely want to avoid S+CRT in the early stage group. We know that S+RT vs RT alone likely have similar if not equivalent survival. The question is: which has a better long-term toxicity profile?  S+50-60GY RT vs 70Gy CRT ? This would be important to comment on.
    5. The RECOMMENDATION for adjuvant RT vs RECEIVING adjuvant RT should also be addressed. Many patient who have indications for RT refuse RT and most do very well. There is a recent paper looking at the trends of adjuvant treatment use in TORS (NCDB analysis) that you should consider commenting on:

      James E Bates  1 , Kathryn E Hitchcock  1 , William M Mendenhall  1 , Peter T Dziegielewski  2 , Robert J Amdur  Comparing national practice versus standard guidelines for the use of adjuvant treatment following robotic surgery for oropharyngeal squamous cell carcinoma Head Neck. 2020 Jun 1. 

      1. "Results: Approximately two-thirds of patients received radiotherapy after robotic surgery for early-intermediate stage oropharyngeal cancer. One in five patients with an indication for adjuvant radiotherapy and 1/3 with an indication for adjuvant chemotherapy did not receive recommended adjuvant therapy."
      2. Interestingly this paper shows that patients who received and those who did not receive adjuvant therapy had no difference in survival. This suggests that surgery alone may be sufficient for many patients. Our traditional indications for adjuvant therapy in p16+ cases may be overcalled as this is a different disease process than p16 neg. - something worth addressing
      3. 2/3 of the patients in this study received RT but only 1/3 in yours. Both are NCDB studies. Can you comment on this please?
  7. Overall great study. The concept of patient selection is very important as we move from survival to a balance between survival and long-term toxicity. 

Reviewer 3 Report

Identification of Clinical and Socioeconomic Predictors of Adjuvant Therapy After Trans-Oral Robotic Surgery in Patients with Oropharyngeal SCC The authors presented a method for Trans-oral robotic surgery (TORS) in oropharynx cancer. This paper describes adjuvant and radiation therapy in TORS-treated 2999 patients and measure the prognostic survival of them within treatment time. This paper has several strong points that I can mention: - The subject of this research is very interesting, and it will make a lots of readers interested and absorbed to this manuscript; - Big cohort of patients in this study makes this paper statistically strong versus smaller cohorts. - The manuscript is made in a good structure and authors managed to mention their points clearly. However, there are several points that must be corrected before the manuscript goes for publication: - A minor point about this paper would concern a discussion for some more conventional methods involving imaging and AI and their connection to this method. - Another point can be related to more explicit description for inclusion and exclusion parameters. A minor editorial comment: Please make your graph more adjusted. They are readable but can be made with higher quality. In overall, the manuscript discusses a very intriguing research. Therefore, I recommend minor revision. Thank you

Round 2

Reviewer 1 Report

I thank the authors for their comprehensive answer on my remarks. The data have been extensively analyzed but unfortunately important data about locoregional control, complications and selection criteria are missing. These extensive shortcomings have been addressed in the discussion but could also be mentioned in the abstract.

In the discussion the authors state:

"However, we were encouraged to see that treatment with TORS
 resulted in the elimination of RT in 33% of patients. For some patients, particularly those with HPV positive and low volume of disease, surgery alone may be adequate and reduce the short- and long-term complications of chemotherapy and/or RT. "

This remark is not supported by the study. Even small HPV positive tumors may have more extensive neck disease with a need for (chemo)radiation. Why not mention that even for small HPV positive tumors in a considerable percentage TORS will have be followed by (chemo) radiation. So, what does TORS really add to (chemo) radiation?
